# The Mediation Effect of Attitudes for the Association between Thoughts and the Use of Condoms in a Mobile-App Environment: From Thought to Intention

**DOI:** 10.3390/ijerph192013631

**Published:** 2022-10-20

**Authors:** Felipe Besoain, Ismael Gallardo

**Affiliations:** 1Faculty of Engineering, Campus Talca, Universidad de Talca, Talca 3460000, Chile; 2Faculty of Psychology, Campus Talca, Universidad de Talca, Talca 3460000, Chile

**Keywords:** mobile devices, attitude change, persuasion, health promotion, behavior, prevention STI

## Abstract

The ubiquity of mobile devices and access to the internet has changed our daily life and, in some cases, promoted and facilitated social and sexual interrelationships. There are many applications of technology and campaigns promoting healthy behaviors and prevention of sexually transmitted infections. Can we develop a strategy for the same purpose using mobile devices, based on the theory of attitude change? We developed an app and tested it with 105 undergraduate students, where they had to actively think in favor of condom use with a high amount of elaboration, leading to attitudes and behavioral intention (BI) in concordance with contemporary theories about attitudes and behavioral change. PROCESS macro models were used to analyze potential mediations. Results show a significant correlation between thoughts and attitudes, and attitudes partially mediated the association between thoughts and condom use. Individuals with positive thoughts tended to positively correlate their thoughts with their attitudes, and, consequently, these attitudes with their BI. In this study, we showed that (1) it was possible to develop and test an app based on the elaboration likelihood model (ELM); (2) consistent with previous studies, attitudes partially mediated the association between thoughts and condom use (BI) in a mobile environment; and (3) applications of this strategy can be used to build new approaches for prevention in health care.

## 1. Introduction

Information and communication technologies (ICT) have facilitated interpersonal relationships, in general, through the Internet and mobile applications. Many studies show how the Internet has been considered as a connection point to meet sexual partners [1,2]. Moreover, since smartphones and tablets were introduced to the market, information has become increasingly omnipresent and, as a result, applications and social networks have become more accessible online [3,4]. Another modern tool for finding sexual partners, both heterosexual and homosexual, are online dating mobile apps [5,6,7]; these apps use the Internet to take into account factors such as timing, location and profile, among others [8]. Furthermore, dating apps have been shown to impact the spread of sexually transmitted infections (STI) for various reasons, including: the ease with which sexual partners can be found, the free cost of the apps, and the wide span of dating apps across both heterosexual and homosexual communities. In this context, it has been shown that people who use dating apps tend to take more risks when they have a sexual encounter [9] such as, for example, not using condoms, among other psychological factors [10].

Currently, there are a number of approaches to affecting beliefs that people have about this type of protection mechanism. For example, encouraging group conversations regarding sexual care improves sexual-care behaviors [11,12], and reviewing videos related to sexual risk behaviors motivates those who observe them to think systematically about their decisions and improves self-efficacy related to condom use, as compared to a baseline and a control group [13,14], among other approaches. Although ICTs are also used, most of them are limited to personal computers (PCs), online videos or written information on websites [15,16], whereas most young people and adults have now migrated from PCs to smartphones, where interventions to promote condom use are not as common. Although mobile apps to promote condom use are available on the market, apps based on theories that predict the ways in which beliefs can be changed, created and, more importantly, predict protective behavior, are very scarce (e.g., [17]). It is also important to note that studies such as [18], though outside the field of public health, have shown mixed effects of the adoption of mobile apps, showing that there are multiple factors involved in the success of an app and user behavior, reinforcing the importance of theory-based work. In this regard, positive interventions on beliefs about condoms using smartphones might not be guided with proper theories about how beliefs change, form or predict behavior, leaving the positive and negative consequences on behavior unattended. For instance, the present research presents a mobile app based on theories on attitude change that predict how and when thoughts lead to behavior, in particular, the elaboration likelihood model of persuasion [19]. The prediction is that when a mobile app promotes a certain type of thinking process, it will lead to positive beliefs about condom use and, consequently, to positive intentions to use them. This is relevant for health organizations and health systems because they work continuously on prevention with limited resources, such as time and budget; in addition, it is important to develop a systematic plan of sexual education at school [20]. Furthermore, condoms are well known as an accessible, available, and low-cost item that efficiently prevents STI.

In this context, it is necessary to establish new strategies that allow the population to consider the use of condoms in a sexual relationship as a viable method to prevent STIs, especially when face-to-face activities might be limited by quarantine or confinement in a pandemic context, which has negatively impacted the use of condoms among young adults [21,22]. For instance, smartphones can serve as new tools to improve protective health behavior, given their daily use, their privacy, and their ability to detect the geographical locations with a greater probability of sexual encounters [13,23,24,25,26,27,28]. Therefore, new preventive actions should focus on personal ICTs such as smartphones.

Smartphones can be used as persuasive technologies, which are interactive systems designed to aid and motivate people to adopt behaviors that are beneficial to them and their community while avoiding harmful ones. The use of these technologies in order to bring about desirable change by shaping and reinforcing behaviors and/or attitudes is growing in virtually all areas of health and wellness, incorporating concepts of social psychology and rhetoric, among others [29].

In this work, we present a mobile app that favors condom use as a protective health behavior based on predictions using basic principles of attitude change to understand when thinking about condom use leads to positive intentions to use condoms. In the following subsections, we will explain both the conceptual and technical bases of the proposal, particularly, how thoughts lead to intentions when participants have positive attitudes toward performing a behavior.

### Thoughts, Attitudes, and Behavioral Intentions

Several investigations in social psychology have consistently shown that thoughts and intentions to behave can be related, especially when those thoughts lead to attitudes toward that behavior [30,31,32,33]. “Thoughts” refers to the cognitive responses that people generate as a response to certain information or an issue [32,33,34] such as, for example, condom use. “Attitudes” refers to evaluations that people hold which can be positive or negative, good or bad, toward objects, people, behaviors or any possible object that is technically named an “attitudinal object” [19]. “Behavioral intentions” refer to a person’s readiness to act, and they are considered as the immediate antecedent of behavior [35]. Attitudes, for instance, are an important part of understanding the relationship between what people think and how they act.

Attitudes are important to human beings because they serve several functions [36]. They allow people to actively search for information stored in memory and, under certain circumstances, they are a behavioral guide; depending on the direction of the evaluation, they lead to approach or avoidance perspectives of social objects (e.g., condom use or misuse), and they serve to express people’s identity.

The importance of attitudes is evident in persuasion research, in which attitudes are changed or formed through different types of information that favor a certain behavior. In this context, when confronted with information related to a behavior, a person can generate “cognitive responses” or “thoughts” regarding the behavior that can be positive or negative [19,33]. Consequently, thoughts lead to judgments and evaluations (i.e., attitudes) about the object that are consistent with thought favorability. In other words, to understand subsequent attitudes and behaviors, the direction of the message is less important than the direction of what people think [19,34,37].

Not only is the direction or favorability of thinking important to understand attitude formation and change, but also the amount of thinking. The more motivated and capable a person is to engage in thinking as an activity (i.e., a large amount of thinking), the more difficult are the resulting attitudes to change, and they persist longer through time than attitudes based on low motivation and capacity to think (or a low amount of thinking) [38,39,40]. For example, one study [41] showed that people who evaluate information in conditions of a high amount of thinking hold their evaluations longer than people who do not think about the information they receive. Another study [42] showed that when people have formed an attitude by making a cognitive effort, it is more difficult for them to change it when faced with counter-attitudinal information than it is when their attitudes are based on a low amount of thinking [43]. These attitudes are not only more difficult to change, but also more stable [44]. Thus, thoughts that are positive in direction and based on a high amount of thinking lead to positive attitudes toward a certain object (e.g., a behavior).

In order to evaluate thought favorability, research tends to use persuasive messages with arguments in a favorable or unfavorable direction, with strong or weak arguments, and with one or two sides, among others (see, for a review, [31,34]). Regarding the amount of thinking, variations (high vs. low) in participants’ issue involvement, personal implication, and, especially, personal relevance, have been shown to impact a person’s motivation to think [34]. Nevertheless, the study of attitudes can strongly benefit from a direct request to generate thoughts, instead of only observing them after receiving new information [45]. Thus, asking people explicitly to think of arguments for or against the use of condoms can generate favorable or unfavorable attitudes, respectively, within a task that explicitly asks people to dedicate cognitive effort. This simple manipulation has been used successfully to control the direction of the thoughts that people have in relation to various subjects such as the body itself [46], anti-stereotypical figures [47], the consumption of vegetables [48] and even about inhabiting other planets [49]. In all this research, the key is to think for or against a certain object to lead to consistent evaluations in that direction. In contrast to receiving information, which is a relatively passive context, asking people explicitly to think of arguments allows thought generation in an active frame. The self-generation of thoughts implies dedication and motivation, involving great cognitive effort to think about condom use.

Finally, behavioral intentions (BI’s) have been identified from different perspectives as a strong predictor of actual behavior [50,51,52]. Regarding condom use, Von Haeften and colleagues showed that more than 70% of people from high-risk populations (e.g., sex workers) who express the intention to always use condoms reported doing so [53]. Although several dimensions can affect BI’s, attitudes are one of the most extensively known to predict further intentions, especially when they are perceived as strong rather than weak. For example, when attitudes toward a candidate are perceived as important (a central feature of a strong attitude [54]), there is a positive relationship between having an attitude and voting for a candidate [55]. Strong attitudes are also related to high-amount-of-thinking conditions, consequently affecting BI’s. Moreno and colleagues ([56], Study 3) found that attitudes toward a job candidate predict intention to defend him/her and hiring intentions, especially when participants perceive high rather than low elaboration of a candidate’s CV.

In summary, going from thought to intention is more likely to occur when a cognitive response (a positive or negative thought) is generated in conditions in which participants think in an active (rather than passive) manner, leading to positive or negative attitudes that are more likely to be strong, given the high amount of thinking that was dedicated. Due to the process, a positive or negative BI is generated. In other words, when cognitive elaboration is high, the effect of thought favorability on intentions to use condoms is mediated by attitudes toward condoms.

Considering the different conditions in which thoughts are more likely to guide intentions (positive thought direction and active thinking), this study presents a mobile app by which thoughts can be generated within a smartphone environment using an active frame, favoring the likelihood to guide intentions to use condoms through affecting attitudes toward condom use.

To analyze the proposed pattern of results, mediation analysis was used. Mediation is a statistical procedure to evaluate potential mechanisms to understand the relations among variables (and also effects on a dependent variable in an experimental design). In this case, we will evaluate if attitude direction can serve as a potential mechanism to explain the association between thought favorability and behavioral intention. The arrows in Figure 1 show direction, in this case, in which thoughts influence attitudes; likewise, attitudes influence behavioral intentions (use of condoms). In other words, once thoughts have their effect on attitudes, attitudes will produce changes in behavioral intentions.

For instance, we hypothesized that: (H1) thoughts would be significantly associated with attitudes, and (H2) attitudes would significantly and partially mediate the association between thoughts and use of condoms (Figure 1).

The paper is structured as follows: first, the methods used to create the application are introduced, with an evaluative study of the application; second, we present the results with Bivariate Pearson correlations and a mediation analysis; finally, we discuss our results and describe future work.

## 2. Materials and Methods

This research follows the design and creation method [57] to develop a mobile application and a non-experimental study to evaluate our theory-driven hypothesis.

### 2.1. Mobile-Application Development

We developed an Android OS application to test our hypothesis on a smartphone (mobile environment). Through this app, users are able to: (1) write self-generated thoughts in favor of condom use, (2) evaluate the messages written, and (3) complete a questionnaire about the intention for condom use and demographics data.

More details about the development, such as the architecture of the solution (scheme with a subdivision of layer models, subdividing it into presentation, domain, and data layers) along with a feasibility test, can be found in [58].

To present and see the principal functionalities of how the application behaves, we present two general-use cases describing how the users and application interact:Writing and evaluating thoughts in favor of condom use.Completing a behavioral intention questionnaire.

For the development of the application, we used the Flutter framework. Once the user installs the application (called Think About It), a short tutorial is shown to the user in three steps: first, it tells them where to write their thoughts (Figure 2 left and center); second, it shows them where to evaluate the thoughts (Figure 2 right); third, it tells them where to answer the intention questionnaire. Steps one and two are related to the use case: “Writing and evaluating thoughts in favor of condom use” (see Table 1), whereas step three is related to the use case: “Completing a behavioral intention questionnaire” (see Table 2).

In every step, the data is stored in a database associated with a unique identifier device (UID) from the smartphone. All the data is anonymous since there is no personal information associated with the collection data. A description of this analysis is included in the statistical analysis section.

### 2.2. Evaluative Study of the Mobile Application

#### 2.2.1. Participants

A total of 105 undergraduate students [59,60] (32 female, 63 male, 2 non binary, 8 no response; age M = 22.6 SD = 4.35) were recruited and asked to download the mobile application. This sample was selected since university students are at a high risk of STI contagion [9]. Completion of the study activity and questionnaire occurred at the end of their classroom lecture and took approximately 20 min.

#### 2.2.2. Procedure

We invited students to participate in research about the development of an application to distribute information on safer sexual behavior. They were informed that their task included writing and evaluating different ideas about condom use, and that they needed to download an app. In this context, all the participants gave informed consent at the beginning of the procedure. This acceptance allowed them to download the mobile application through a QR code.

Once the app was downloaded, participants had to write their ideas favoring condom use through their smartphone keyboard. They were asked to write as many ideas as possible and, after finishing, press a “save idea” button to start writing a new one. When participants stopped writing, they moved on to a new screen in which each idea was presented separately in order to be qualified in terms of how favorable, persuasive, or convincing it was. Then, participants had to evaluate their attitudes and intentions to use condoms through a series of items. Finally, demographics were presented, and a final message thanked them for their participation. The protocols for compliance with ethical requirements were approved by the ethics committee of the University of Talca (ID: 15-2021).

### 2.3. Instruments

#### 2.3.1. Thought Favorability

Once the participants wrote down their thoughts (ideas), they had to evaluate each of them in terms of how persuasive (unpersuasive), convincing (unconvincing) and favorable (unfavorable) they were in a series of 10-point semantic differential items (−50 to 50, including 0 as neutral). The items were adapted from previous research [49]. Given the high correlation among items (α = 0.815), we averaged them to create a single index of thought favorability. Higher values indicate more favorable thoughts.

#### 2.3.2. Attitudes toward Condom Use

To evaluate attitudes towards the use of condoms, 4 scales of 7-point semantic differentials (1–7) were used (e.g., good–bad, useful–useless, necessary–unnecessary, and recommended–not recommended). The items were adapted from previous research [30]. Given their high internal correlation (α = 0.844), we averaged them to create a single index of attitudes. Higher values indicate more favorable attitudes.

#### 2.3.3. Behavioral Intentions to Use Condoms

BI to use condoms were assessed with four items, adapted from a previous study [61], on a seven point Likert scale from “very unlikely” to “very likely”: “If you were going to have sex in the next three months, how likely is it that every time you have sex you will actually use a condom?”; “If you were going to have sex in the next three months, how likely or unlikely is that in every time you have sex you will use a condom even if your partner does NOT want to?”; “If you were going to have sex in the next three months, how likely or unlikely is it that every time you have sex, you will tell your partner that you need to use a condom?", and “If you were going to have sex in the next three months, how likely or unlikely is it that every time you have sex, you will discuss safer sex with your partner?" Small wording modifications were made from the original study, with the aim of adapting the items from English to Spanish. Although the original work indicates a two-factor scale (item 1 and 2 for condom use intentions, and 3 and 4 for sexual communication), an exploratory factor analysis showed that the four items represent only one dimension. Therefore, a single behavioral intention index was created by averaging them (α = 0.895). Higher values indicates positive behavioral intentions to use condoms.

#### 2.3.4. Statistical Analysis

To evaluate the statistical significance of the mediational trends, a bootstrapping procedure [62] was used, which treated the data pattern obtained as the population, extracting randomly (through substitution), five thousand samples. The estimations of the indirect effects on BI’s were calculated for every sample to which the procedure was applied, to generate an interval of confidence (CI) for the indirect effect (pattern ab, Figure 1). If the value “zero” was found outside of the CI, it was indicative of the indirect effect of attitude toward condom use on the relation between thought favorability and BI’s.

## 3. Results

### 3.1. Descriptive Analyses

Bivariate Pearson correlations were performed between the different variables. As observed in Table 3, thoughts (M = 25.26, SD = 23.93), attitudes (M = 6.34, SD = 1.07) and intentions to use condoms (M = 5.33, SD = 2.15) are positive and significantly related.

### 3.2. Mediation Analyses

To describe mediation, unstandardized regression coefficients are presented. As shown in Figure 3, thought favorability influences behavioral intention indirectly through the effect of attitudes (ab = 0.0086, 95% bootstrap CI = 0.0000 to 0.0225 ). Users with more positive thinking report more positive attitudes (a = 0.0109), and these more positive attitudes are associated with greater behavioral intention (b = 0.7853). From a different perspective, the results show that positive thinking led to positive BIs (i.e., the total effect, c), but the relation between them is due to attitudes in response to positive thinking (i.e., the indirect effect, ab).

In concordance with previous results considering the BI separated into two dimensions, condom use intention and sexual communication index, results show that attitudes mediate the relationship between thoughts and BI (considering the BI, separated into two dimensions, condom use intention and sexual communication index, similar trends are observed. The total effect for condom use intention is (c = 0.0291, se = 0.0084, *p* = 0.0008, CI = 0.0125 to 0.0457) and the indirect effect is (ab = 0.0092, se = 0.0060, CI = 0.0001 to 0.0235). On the other hand, the mediation model with the sexual communication index has a result of a total effect for sexual communication of (c = 0.0353, se = 0.0076, *p* = 0.0000, CI = 0.0203 to 0.0503) and the indirect effect is (ab = 0.0080, se = 0.0055, CI = 0.0001 to 0.0210). In this context, the results show that positive thinking led to positive BI’s (i.e., the total effect, c), but the relation among them is due to attitudes in response to positive thinking (i.e., the indirect effect, ab).).

## 4. Discussion

In this study, we evaluated the impact of the generation of favorable thoughts toward condom use on users’ attitudes and BI toward condom use. However, most laboratory and field studies have been conducted in non-technological contexts and little is known about their application in mobile-phone environments. Measurements of thought favorability, attitudes and BI were collected to evaluate when participants moved from thought to intention.

As hypothesized, thought favorability is significant and positively related to attitudes toward condom use (H1), and attitudes are significant and positively related to intentions to use condoms (H2). As a consequence, and as Figure 3 shows, there was an indirect effect of attitudes on the relationship between thoughts and BI, suggesting that attitudes toward condom use (i.e., positive or negative evaluations) are important elements to predict positive and negative BI’s to perform healthy behaviors. In other words, the module explored on Think About It can help promote condom use as a healthy sexual behavior.

The results of our study show that Think About It can replicate the theory-driven predictions by allowing participants to write what they think about the issue using their own words and meaning. According to the ELM, positive thoughts and positive intentions will be related through attitudes toward the conduct, especially when participants actively think about the behavior to perform [19]. In our work, asking participants to write an idea or argument in favor of condom use enhanced motivation and capacity to think, establishing a theoretical condition of high cognitive elaboration, allowing positive and significant correlations between the resulting attitudes and behavioral intentions toward condom use. This prediction is supported by mediation analysis, which strongly suggests that relations between thoughts and intentions are linked through attitudes.

It is important to note that these processes of high elaboration activate different dimensions of an attitudinal object, because individuals consider more characteristics or elements. For instance, not only is condom use affected but also sex-related communication. However, attitudes toward pregnancy, alternative STI prevention methods or STI testing can also be affected [63]. Moreover, high elaboration processes enhance the likelihood of the person evaluating alternative attitudinal objects and behaviors related to condom use. Thus, smartphone interventions for condom use based on high-thinking conditions can positively affect different dimensions of the object, allowing more stability of behaviors and attitudes through time.

Another important issue is the procedure that Think About It uses to promote positive relationships among variables. As a positive correlation indicates, positive values in one variable are related to positive values in the other, but negative values too. Nevertheless, as the descriptive values indicate, Think About It tends to generate positive values either on thoughts, attitudes or behavioral intentions. This is because the procedure instructed to think only in one direction (i.e., promote a protective behavior). This is very important, because the absence of that instruction may increase the likelihood of thinking about either positive or negative elements of using condoms in similar quantities or even negative thoughts about condoms. In the first case, thoughts may lead to positive and negative evaluations simultaneously, leading to attitude uncertainty, which reduces the impact on BI’s [37]. In the second case, by generating mostly negative thoughts, attitudes toward condom use may also be negative and might predict unhealthy behaviors such as avoiding the use of condoms. It is also important to point out that, in some exceptional cases, the instruction of writing a positive sentence may cause the person to develop negative attitudes. This may happen, for example, when users have strong and stable negative beliefs toward condom use and the instruction of positively biased thought make salient the negative belief. An improvement in this direction should include the incorporation of these potential conditions as initial measurements and create alternative technological procedures to reduce their impact.

## 5. Limitations and Future Research

The results present strong evidence that a theory-based app led to favorable intentions to use condoms. However, it has some limitations. Particularly, given the non-experimental design, it is impossible to know if participants generate positive thoughts as a result of the induction or simply because any thought generated in a participant’s phone tends to be positive. That situation can be ruled out by controlling the type of device used (comparing to a PC) and by providing a negative thought toward the condoms condition (Nevertheless, special attention is needed in this regard. A pure negative thought condition could polarize initial negative thoughts or persuade participants to behave in an unwanted direction). Future research should take into account those elements in order to compare alternative explanations for the results.

Smartphones have an important potential given the fact that they are popular, personal and that they are part of our daily routine. People engage in interactive activities with their smartphone through apps, games, and other software on a daily basis. In this context, it is important to consider these results for future health interventions based on theory through smartphones, increasing the likelihood of success for the use of condoms and other attitudinal objects using mobile devices with this approach.

First, Think About It can be technologically modified to evaluate different objects or promote healthy behaviors in areas such as eating practices, sports, or nutrition [23,27]. It can also be modified to incorporate alternative strategies that literature has shown to be positively associated with BI’s, such as avatars or gamification. For example, Peña and colleagues [64] have reported that participants assigned to obese instead of thin avatars demonstrated less physical activity while playing a virtual tennis game in an exergame [64]. Besoain and colleagues [26] use gamification (among other variables) to promote healthy sexual behavior among men who have sex with men (MSM) in a smartphone environment. Second, regarding theory-based interventions, results might be different depending on the type of display used to promote changes. Smartphones or head-mounted displays (HMD) can be very similar in the procedure used to generate thoughts but include variations in different dimensions of virtuality such as presence or immersion, leading to different consequences [65,66]. Likewise, psychological processes in which thoughts are more likely to be used by the users [67] may also be possible to incorporate by changing technical elements, thereby improving potential health indicators.

Finally, we believe that studying contemporary attitude change theory in mobile and technological environments will help to build new approaches for prevention in health care. Users respond differently to the reception of information, for instance, healthy messages. The more the information is processed, the greater the likelihood of creating strong attitudes, and, with this, the ability to predict behavior. Therefore, it is important to not only use technology to distribute or present information, but also to understand how it is processed by people.

## Figures and Tables

**Figure 1 ijerph-19-13631-f001:**
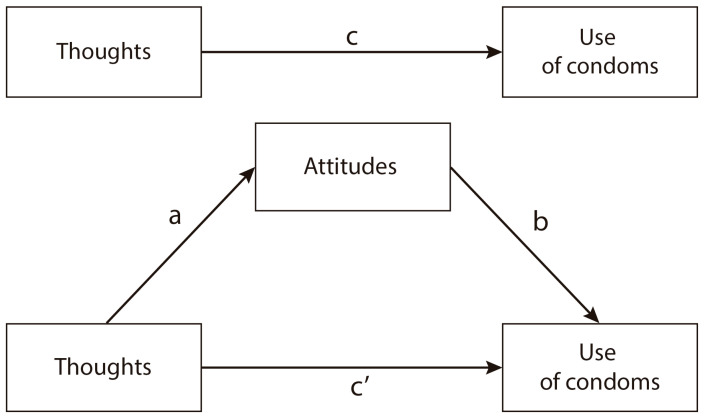
Hypothesis mediation model for thoughts, attitudes, and use of condoms. Thoughts: predictor variable, attitudes: mediator, use of condoms: outcome variables.

**Figure 2 ijerph-19-13631-f002:**
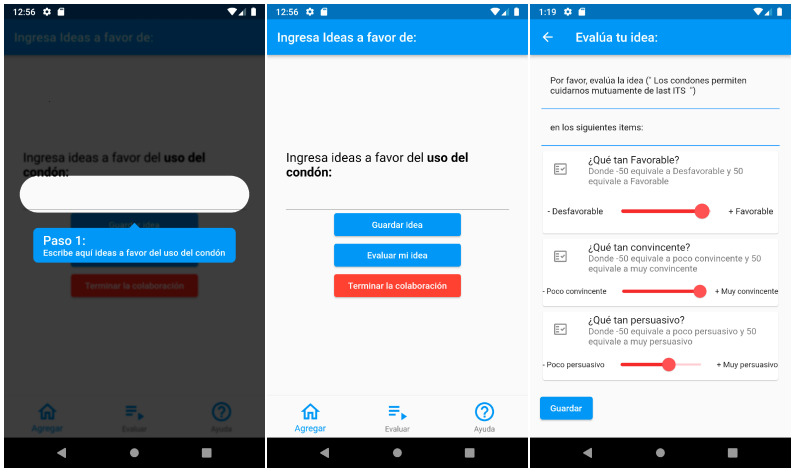
Mobile application: Think About It. **Left**, Tutorial that shows users what to do in the application (*Translation from Spanish to English of blue overlay: Step 1: Write here ideas in favor of condom use.*). **Center**, Graphical interface where the user can write an idea or argument in favor of the use of condoms (*Translation from Spanish to English: Write ideas in favor of condom use: Buttons from top to bottom: (blue button) Save Idea, (blue button) Evaluate my idea, (Red button) Finish the collaboration.*). **Right**, Evaluation of the idea or argument (*Translation from Spanish to English: From top to bottom, please evaluate the idea (“text of the idea to be evaluated”); In the following items: How favorable? How convincing? How persuasive? (blue button) Save*).

**Figure 3 ijerph-19-13631-f003:**
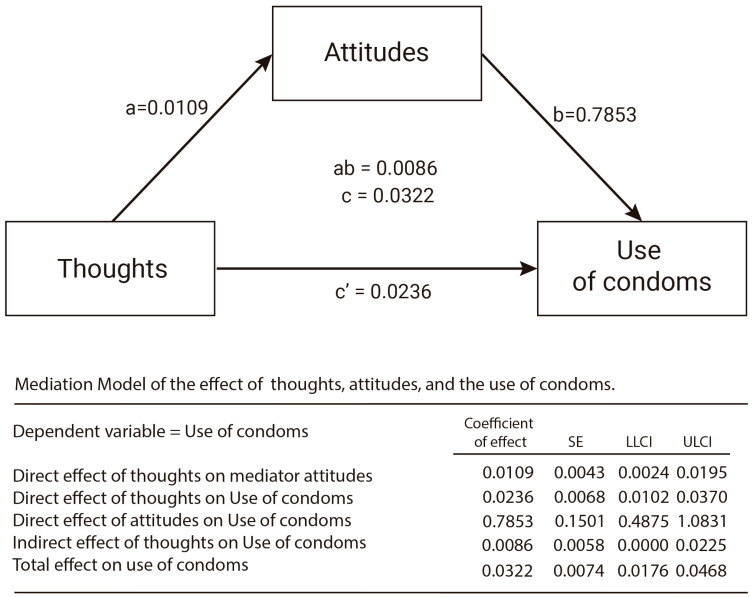
Mediation model for thoughts, attitudes, and use of condoms. Thoughts: predictor variable, attitudes: mediator, use of condoms: outcome variables (a = unstandardized coefficient of thoughts on attitudes; b = unstandardized coefficient of attitudes on use of condoms; ab = indirect effect of thoughts on use of condoms; c′ = direct effect of thoughts on use of condoms; c = total effect of thoughts on use of condoms = ab + c′).

**Table 1 ijerph-19-13631-t001:** General use case: writing and evaluating thoughts in favor of condom use.

Use case:	Writing and evaluating thoughts in favor of condom use.
Actor:	User
Purpose:	Allow the user to write and evaluate thoughts on the application.
Summary:	This use case begins when users open the application and the main screen prompts them to write a thought in favor of condom use. Users can add as many thoughts as they want. Then, they have to evaluate each one of them (how favorable, persuasive, etc., they are).
Preconditions:	(1) To evaluate a thought, the user must first write it.

**Table 2 ijerph-19-13631-t002:** General use case: completing a behavioral intention questionnaire.

Use Case:	Completing a behavioral intention questionnaire.
Actor:	User
Purpose:	Present a behavioral intention questionnaire to the user.
Summary:	This use case begins when users finish evaluating their thoughts, which initiates a questionnaire about intentions and demographic data. After the users finish answering the questions, the app thanks them and tells them they can uninstall the application.
Preconditions:	(1) The user must have evaluated all the thoughts on the application.

**Table 3 ijerph-19-13631-t003:** Correlations among study variables.

	1	2	3	4	5
1. Thoughts	-	0.243 *	0.396 **	0.324 **	0.417 **
2. Attitudes		-	0.506 **	0.474	0.468 **
3. Behavioral intention			-	0.936 **	0.927 **
4. Condom use intention				-	0.737 **
5. Sexual communication					-

* the correlation is significant at *p* < 0.05; ** the correlation is significant at *p* < 0.01.

## Data Availability

Not applicable.

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
