# Peer review of "The Mediation Effect of Attitudes for the Association between Thoughts and the Use of Condoms in a Mobile-App Environment: From Thought to Intention"

_ijerph, 2022, doi:10.3390/ijerph192013631_

Round 1

Reviewer 1 Report

Dear Authors

It is great to read your interesting article.

I provide some comments that may help you to improve your manuscript:

 The abstract structure is acceptable, and the paper title is very informative and relevant to the content. It contains most of the required details, including the aim, and a summary of the main findings. However, I recommend the authors to add one line indicating method summary and research implications.

The background in the introduction could be improved by adding more related and recent reference. In addition, it is better to add a paragraph at the end of the introduction which saying about the structure of your paper.

The paper lacks a separate literature review part, which may provide a theoretical framework with regard to the research gap and hypotheses development; literature review should be related to the study field - it should be added to the paper as a separate section after the introduction The author(s) should better to be deeper in scanning the related studies and expand their work.

It would better to more clarifying the population from which the participants were selected and why this population was considered? and if the number of respondents received are enough?

Statistical analysis and Mediation analyses in page. 7 need more explanation.

I advise the authors to bring some notes about the consideration of research generalization.

The paper suffers from clear contribution in the field and has some weakness in the limitations, future research and implications. The authors are suggested to better assess their own work and the impact of the research, followed by the importance of the results, both theoretically and practically.

Finally, the last part (Abbreviations) lack and limited to some of them, kindly double chick for example (HMD) page 9 line 316. Additionally, references need to add more recent references related to the research topic.

Bests of luck

Reviewer 2 Report

This is a very meaningful study, which presented an app and tested a strategy using mobile devices based on the theory of attitude change with 105 undergraduate students. In some sense, these results show the mediation effect of attitudes for the association between thoughts and the use of condoms in a mobile app environment. This discovery is valuable for health promotion. More new literature on mobile app needs to be cited and introduced, such as "The Dark Side of Mobile App Adoption: Examining the Impact on Customers' Multichannel Purchase", "Assessing Mental Health among College Students Using Mobile Apps: Acceptability and Feasibility", "PRO-VAS: utilizing AR and VSLAM for mobile apps development in visualizing objects", "Measuring Ease of Use of Mobile Applications in E-commerce Retailing from the Perspective of Consumer Online Shopping Behaviour Patterns", and "Branded Apps and Their Impact on Firm Value: A Design Perspective".

Reviewer 3 Report

Authors present results derived from an experimental evaluation that aims at investigating the effects of favorable thoughts concerning condom use on users' attitudes and behavioral intention with respect to condom use. The topic is an important issue in Public Health and is relevant to IJERPH. Some comments and suggestions for improving the final version of the authors' article are listed as follows. 

1) Section 1 is unbalanced since only a few paragraphs are related to the study performed by the authors (e.g. the last paragraph of Section 1 that is immediately before Section 1.1). 

2) Translate the text that appears in Figure 2 since the article is written in English.

3) Regarding the use cases presented, I missed the adoption of a formalism to represent them. There is no formal definition about the use cases reported. No formalism has been used to illustrate them.

4) With respect to the hypotheses tested, I would suggest the inclusion of further details. For example, what is the null hypothesis of each of them? Also, in the discussion section, a single comment is given for both hypotheses. If this is done, then why the proposal of two hypothesis is made? Additional information about the goals of the hypotheses test is needed.

5) The addressed topic is important and timely. However, interesting results derived from a descriptive analysis are missing. The proposed study is an exploratory approach and deserves attention if the results make sense. Thus, some graphs built with the collected data and aiming at describing, showing, or summarizing some data aspects in a constructive way, such that patterns might emerge, would be helpful.

Round 2

Reviewer 1 Report

Thank you for addressing all my suggested comments.

Best of luck